# *Pityopsis ruthii*: An Updated Review of Conservation Efforts for an Endangered Plant

**DOI:** 10.3390/plants12142693

**Published:** 2023-07-19

**Authors:** Phillip A. Wadl, Adam J. Dattilo, Geoff Call, Denita Hadziabdic, Robert N. Trigiano

**Affiliations:** 1U.S. Vegetable Laboratory, United States Department of Agriculture, Agricultural Research Service, Charleston, SC 29414, USA; 2Tennessee Valley Authority, Knoxville, TN 37902, USA; ajdattilo@tva.gov; 3United States Fish and Wildlife Service, Cookeville, TN 38501, USA; geoff_call@fws.gov; 4Department of Entomology and Plant Pathology, The University of Tennessee, 2505 EJ Chapman Drive, Knoxville, TN 37996, USA; dhadziab@utk.edu

**Keywords:** Ruth’s golden aster, Asteraceae, plant conservation, endangered species, ripraian

## Abstract

*Pityopsis ruthii* (Small) Small, Ruth’s golden aster, is an endangered Asteraceae species that grows in the riparian zone along small sections of two rivers in the Southern Appalachian Mountains of the United States of America (USA). Since 1985, the species has been listed under the Endangered Species Act by the United States Fish and Wildlife Service (USFWS). The mission of the USFWS is to conserve, protect, and enhance fish, wildlife, and plants and their habitats for the continued benefit of the American people. The agency provides national leadership in the recovery and conservation of imperiled plant species by working with the scientific community to protect important habitats, increase species’ populations, and identify and reduce threats to species survival with the goal of removal from federal protection. Over the past 35 years, research efforts have focused on studies designed to delineate the range and size of populations, determine habitat requirements, reproductive and propagation potential, and understand the demographic, ecological, and genetic factors that may increase vulnerability to extinction for *P. ruthii*. Cooperative partnerships have driven the completion of actions called for in the strategy to recover *P. ruthii*, and in this review, we highlight these efforts within the context of species conservation.

## 1. Introduction

Anthropogenic activities [1], including those that induce climate change, have had profound negative effects on populations and distributions of plant species across the globe [2,3,4], and in 2017, the extinction of thousands of species worldwide was predicted [5]. Declines or extinctions of species due to climate changes could be exacerbated by the loss or limitations of appropriate habitats [6], competition among species occupying similar niches [7], competition for pollinators [8], and low seed dispersal and success rates, among others [9]. However, the increase in maximum annual temperatures appears to be a principal driving force associated with many local extinctions along with, to a lesser extent, changes in precipitation variables [3].

In the USA, over 900 plant species have been listed under the Endangered Species Act (ESA) and the U.S. Fish and Wildlife Service (USFWS) oversees the protection of these species. The mission of the USFWS is to conserve, protect, and enhance fish, wildlife, and plants and their habitats for the continued benefit of the American people. The agency provides national leadership focused on the recovery and conservation of imperiled plant species. This is accomplished through coordination with the scientific community to protect important habitats, increase species’ population sizes, and identify and reduce threats to species survival with the goal of removal from federal protection. All federally listed endangered species in the USA are required by law to have a recovery plan developed by the USFWS. These recovery plans provide a framework with detailed specific management actions for private, Federal, and State cooperation for conserving listed species and their ecosystems. Although this is a non-regulatory document, it provides guidance on how best to achieve species recovery and/or protection.

## 2. *Pityopsis ruthii*

*Pityopsis ruthii*, or Ruth’s golden aster, is a federally endangered herbaceous perennial plant that has been protected by the ESA since 1985 in the USA [10]. This species grows only in highly restricted, very short, stretches of the Ocoee and Hiwassee River systems in Southeastern Tennessee (Figure 1) [11]. Factors regulating the small geographical distribution and narrow habitat of *P. ruthii* are not well-explored and are poorly understood. The Ocoee and Hiwassee rivers are in close proximity (approx. 15 km) but are separated by mountainous terrain. This spatial distribution created potential limitations for the exchange of genetic material and possible isolation of the populations from both rivers. Over the past 35 years, research efforts have focused on studies designed to delineate the range and size of populations, determine habitat requirements, reproductive and propagation potential, and understand the demographic, ecological, and genetic factors that may increase vulnerability to extinction for *P. ruthii*. In the case of *P. ruthii*, the recovery plan outlines five actions that are needed for recovering the species and assuring that viable, self-sustaining populations exist on the Hiwassee and Ocoee rivers [12]. These actions are the following: (1) determine asexual and sexual reproductive biology, (2) determine habitat requirements, (3) obtain life history, (4) define what constitutes a viable population, and (5) determine and implement management actions needed to ensure the continued existence of self-sustaining populations on both rivers. This mini-review focuses on the research progress over the past 35 years within the context of meeting the objectives of the recovery plan (Table 1). Here, we chronicle research findings over the past 35 years in relation to the biology, genetics, and conservation efforts of *P. ruthii*.

## 3. Species Description and Systematics

*Pityopsis ruthii* was discovered by Albert Ruth in the Hiwassee River valley and the species was described first as *Chrysopsis ruthii* by John Small [14]. The species is a herbaceous, tufted perennial plant with slender stoloniferous rhizomes that is up to 30 cm in height with a plant habit described as erect to ascending or decumbent, stiffish, and cylindrical or slightly tapering without substantial furrows or ridges (terete). Lanceolate leaves (2–5 cm long) are silvery pubescent and overlap in tight spirals; inflorescences are either solitary or in a cyme (1.5–2 cm diameter heads) composed of 8–15 ray florets, with bright yellow petals and numerous disk florets producing ~4 mm long pubescent achenes (seeds) each with a 4–5 mm pappus (Figure 2A,B). Flowering occurs from July to frost and peak flowering is in September. The species is also characterized by habitat specialization, as distribution is restricted to discrete locations on exposed phyllite boulders within and along the banks of the Hiwassee and Ocoee rivers (Figure 2C) [15]. The taxonomic classification of Ruth’s golden aster has changed from *C. ruthii* to *Heterotheca ruthii* [16] and finally to *P. ruthii* [17,18,19]. Additional studies have maintained this taxonomic classification [20,21,22,23].

## 4. Genetic Diversity Studies

It has been hypothesized that inbreeding depression or limitation of genetically compatible individuals impacts sexual reproduction both spatially and temporally in *P. ruthii* [24,25]. Knowledge of genetic diversity, population structure, and gene flow is critical for the delineation of what constitutes a viable population and in completing Recovery Tasks 6.3 and 7.4 (determine whether Hiwassee and Ocoee River populations, respectively, are self-sustaining).

The first effort to determine what constitutes a self-sustaining population using population genetic analyses was conducted in 1994 [26]. Analyses of plants from discrete locations on the Hiwassee and Ocoee rivers with two polymorphic allozyme markers found that 84% of the genetic differentiation was within occurrence (sampling location), 15% of differentiation was between occurrences, and 1% was between river systems [26]. Further investigation of chloroplast variation revealed the presence of three haplotypes within plants from the discrete locations: two associated with the upper and lower regions of the Hiwassee River and one associated with the Ocoee River [27]. To further study the genetic diversity and population genetics of *P. ruthii*, Wadl et al. [28] developed 12 nuclear and 5 chloroplast microsatellite markers. Preliminary analyses of individuals from two discrete Hiwassee River locations and one discrete Ocoee River location with these markers agreed with the findings of Sloan [26]; 90% of genetic differentiation was within occurrence and the individuals on the Hiwassee and Ocoee River should be defined as different populations.

By 2011, an exhaustive census and delineation of all known occurrences (discrete locations) of *P. ruthii* was completed [11]. The Hiwassee River population consisted of ~12,000 individuals across 57 discrete locations and the Ocoee River population consisted of ~1000 individuals across 9 discrete locations [11]. Subsequently, the population structure and genetic diversity were assessed for 814 individuals from 33 discrete locations using 7 chloroplast and 12 nuclear microsatellite markers [13]. Higher levels of gene flow and lower levels of population differentiation were discovered for the Hiwassee River when compared to the Ocoee River [13]. The authors recommended the management of the species using a framework of two-to-four subpopulations along the Hiwassee River and two-to-three subpopulations along the Ocoee River. This should guide the selection of seeds or individuals used for augmentation, reintroduction, and/or translocation should these actions be needed for the species’ conservation.

More recently, the genetic diversity of half-to full- siblings of *P. ruthii* from one Hiwassee River population and three Ocoee River populations (*n* = 170) and four populations of *P. graminifolia* (Michx.) Nutt. var. *latifolia* (Fernald) Semple & F.D.Bowers (*n* = 148) were characterized using nine microsatellite markers [29]. *P. graminifolia* var. *latifolia* is a herbaceous perennial that is widely distributed throughout the southeastern USA that grows in close proximity to *P. ruthii*. Genetic diversity estimates were the highest for the *P. graminifolia* var. *latifolia* populations as assessed by the number of alleles per locus and Nei’s diversity index. For the *P. ruthii* populations, these same diversity estimates were the highest in the Hiwassee River population compared to the three Ocoee River populations. Discriminant analysis of principal components using the multiallelic genetic data for the *P. ruthii* populations clearly differentiated the Hiwassee River populations from the Ocoee River populations. This supports the finding of the previous population genetics studies that the Hiwassee and Ocoee rivers should be managed as separate populations. Furthermore, Boyd et al. [29] provided evidence for two genetic subpopulations on the Ocoee River and these results support the recommendation of Hatmaker et al. [13] suggesting that the Ocoee River should be managed as two-to-three viable breeding subpopulations.

## 5. Propagation Methods

The ability to generate plant materials of *P. ruthii* through asexual and sexual reproduction is critical for ex situ and in situ conservation of the species [30,31] and ultimately supports augmentation, reintroduction, and/or translocation efforts should they be needed to maintain viable, self-sustaining subpopulations on the Hiwassee and Ocoee rivers. There was minimal knowledge of the sexual reproduction of *P. ruthii* prior to the publication of the recovery plan. Effective propagation methods are critical for completing Recovery Tasks 3.2 (determine life history, seed germination, and seedling establishment requirements), 6.4 and 7.5 (establish on unoccupied suitable habitat), and 6.5 and 7.6 (establish cultivated populations of plants from the Hiwassee and Ocoee Rivers and provide for long-term seed storage).

Sexual reproduction is required for the establishment of new plants within existing populations and the findings of White [32] indicated that a reproductive barrier was not present in the species as viable seed production was similar in wild-grown vs. greenhouse-grown plants obtained from the Hiwassee River. Although viable seeds are produced, White [32] observed no seedlings in the field and concluded that *P. ruthii* has difficulty establishing on the phyllite boulders. Farmer [33] investigated seed propagation of the species for nursery production as a means toward the expansion of natural populations. Seed heads were collected from plants in late September and dried at ambient laboratory conditions for 24 h. Visual examination indicated that ~5% of the seed were viable. Germination experiments investigating the effect of light and temperature were conducted, and the results indicated that *P. ruthii* seeds had no chilling or special light requirements. Rather, seeds germinate more completely and rapidly at low-to-moderate temperatures (7–24 °C) than at 24–29 °C. Seed germination in the field has been observed between November and January [34], but the seedlings often fail to persist [24,25,32,34,35]. Multiple studies observed pollinators within *P. ruthii* populations but provide no information for identification of the species responsible [14,23,34]. Variable seed set was observed between Hiwassee and Ocoee River populations, with the Ocoee River populations exhibiting higher rates of seed set and seed viability as assessed by germination tests [24]. Boyd et al. [29] collected viable seeds from one Hiwassee River population and three Ocoee River populations and reported highly successful germination from all populations, although germination rates were not provided in their report.

The wild collected seed germinated and grown in a common garden setting produces a dense mat and produce viable seed, but viability declined to 75% within 6 months in seeds that were stored in glass containers at 3 °C [33]. Wadl et al. [25] tested the viability of seeds from long-term storage and germination ranged from 0 to 38% and they attributed the low germination rate to the effects of long-term storage or variability inherent in the collected seeds. There is no way to determine this because there are no records for baseline germination rates or treatment of seeds prior to long-term storage. Regardless, it appears that seed viability is highly variable between river systems and the year of collection.

Production of low numbers of viable seeds is a barrier to the establishment of new plants. Information on the impact of pollinators on seed propagation was absent until recently. To better understand the highly variable sexual reproductive capacity of *P. ruthii*, Moore et al. [36] assessed which insect species may be contributing roles as potential pollinators at in situ and ex situ locations. Species of the Halictidae were common at the ex situ locations and infrequent at the in situ locations. Honeybees (*Apis mellifera* L.) and *Bombus impatiens* Cresson were commonly observed at the in situ locations. The most abundant floral visitor was *Toxomerus geminatus* Say but very little pollen was carried and no pollen was carried by lepidopteran species. Seeds were collected from three Hiwassee River populations and low germination rates were observed for viable seeds. Seeds were considered viable as determined by staining with 2,3,5-triphenyl tetrazolium chloride and germination was defined as successful if the seed produced a green cotyledon. Evidence of inbreeding depression was found as seed viability and germination were higher in controlled crosses made between geographically separated, but genetically similar populations compared to crosses of individuals within a population [36].

Asexual reproduction through stem regeneration of the subaerial root–rhizome crown has been reported and was speculated as the primary method of reproduction for *P. ruthii* [32,37]. A methodology for asexual propagation of the species has been developed recently [25,38]. The in vitro regeneration of plants from flower receptacles and leaf tissue cultured on Murashige and Skoog [39] tissue culture medium supplemented with growth regulators was the first report of asexual propagation for the species [38]. A method for rapid in vitro multiplication of new plants derived from lateral shoots has been developed [25]. Furthermore, Wadl et al. [25] optimized surface sterilization methods to demonstrate the feasibility of in vitro seed germination and developed simple and robust methods for rapid vegetative propagation of terminal stem cuttings. These methods provide a foundation for providing a disease and insect-free source of propagules for use in recovery efforts or germplasm conservation. The long-term limitation of reliance on asexual propagation solely for population development is that clonal propagation of specific individuals alone, without propagation of many individuals, cannot sustain a robust and genetically variable population. However, mass asexual reproduction of many individuals coupled with the development of successful breeding and seed germination schemes can establish a diverse population.

## 6. Conclusions: Recovery Implementation and Future Directions

The life history strategy and strict habitat specialization of *P. ruthii* suggest that the species will always be geographically restricted, but since the species was formally listed as endangered by the USFWS in 1985 [10], a suite of monitoring and research efforts have focused on its conservation. The most impactful of these conservation actions was linked to tasks outlined in the recovery plan for the species [12]. As the primary agency responsible for the recovery of species listed under the ESA, the USFWS uses the best available science to delineate specific tasks that collectively outline a conservation strategy that, if implemented, will help ensure the species is no longer at risk of extinction. The strategy for recovering *P. ruthii* consists broadly of actions intended to improve understanding of the species’ distribution, ecology, and life history; identify positive and negative influences on the viability of populations; develop approaches for managing habitat and populations; and monitor the status of populations in the Ocoee and Hiwassee rivers.

Cooperative partnerships have driven the completion of actions called for in the strategy to recover *P. ruthii*, with contributions from the Tennessee Valley Authority (TVA), USDA Forest Service, Tennessee State Parks, USFWS, academia, botanical gardens, and private industry. The first five-year review of *P. ruthii* [40] summarized the results of several projects that were completed during the 1990s and early 2000s to implement tasks described in the species’ recovery plan. Beginning around 2010, researchers began a more recent phase of recovery projects to better understand the ecology of *P. ruthii* and address ongoing and potential future threats to the species [10,13,25,28,29,38,41,42,43,44]. These efforts focused on multiple topics including completion of on-the-ground delineation of populations along the Hiwassee River, evaluating propagation and reintroduction techniques, permanent monitoring, pests and pathogens, pollination ecology, and population genetics. While results from these efforts have advanced species recovery in new ways, areas remain where future work is needed. The most pertinent unresolved issues surrounding recovery, including the need for improving understanding of how plant releases in the Hiwassee River location affect occupied habitat, and developing strategies to manage flows beneficially for *P. ruthii*, are discussed below.

The TVA operates dams upstream of *P. ruthii* populations on both the Hiwassee and Ocoee Rivers. Since the dam closure, the hydrology of these systems has been altered to support the two primary goals of flood control and the generation of electricity. This has resulted in fundamental changes in habitat for a species that occupies a narrow ecological niche—shallow soils within cracks in rock outcrops that occur in a frequently disturbed zone between the river channel and the surrounding forest. Plants require sunny conditions to establish and reproduce, but cannot tolerate frequent, prolonged inundation with water. While it has long been understood that frequent fluctuations in river flow historically served to scour flooded areas of trees that could shade- out *P. ruthii* [12,15,32,37,40,45], the role of cyclical drought in maintaining open habitat for the species had not been highlighted until Moore et al. [11].

This new understanding that river flows were not the sole mechanism for maintaining all habitats supporting substantial numbers *P. ruthii* plants was encouraging. It suggested that at least some sites were more resistant to the threat of encroaching woody vegetation resulting from altered hydrology, presumably lowering the risk of extirpation for the Hiwassee River population. However, this does not explain how the river interacts with plants at sites where flow appears to be the primary ecological mechanism maintaining habitat for *P. ruthii*.

Determining how flows of varying magnitude, duration, and frequency on both the Hiwassee and Ocoee rivers interact with plants in occupied habitats may be the single most important data gap in understanding the ecology of *P ruthii* and making future conservation decisions. Understanding river flows and how they drive *P. ruthii* population dynamics features prominently in the recovery plan and intersects several specific tasks [12]. River flows of an appropriate magnitude, duration, and frequency are needed for maintaining suitable habitat by eliminating competing vegetation, dispersing seed, and depositing soil in bedrock crevices where *P. ruthii* grows.

Long-term monitoring data provide insight into how future studies could be conducted to determine optimum and operationally achievable flow regimes that would benefit *P. ruthii.* TVA began an annual census on the Ocoee River in 1987, just a few years after entering into an agreement with the state of Tennessee to provide recreational releases of water for up to 116 days per year along the portion of the river supporting *P. ruthii*. Since that time, the total population along the Ocoee River has increased demonstrably from a low of 523 individuals in 1993 to 1388 individuals in 2022; the population has been greater than 1000 plants every year since 2008 [11,46]. Plants have not expanded their distribution along the Ocoee River since 1987 because habitat is inherently limited, but there are more plants in areas where the species has occurred since monitoring began.

The collection of comprehensive population census data on the Hiwassee River began in 2011. While many of the sites on the Hiwassee River appear relatively stable, others have been declining for many years [11]. Sites kept open by drought cycles appear more stable than sites that rely on high-flow water events. Unlike the Ocoee River, the Hiwassee River does not have regular recreational flows and only receives a 25 cubic feet per second (cfs) base flow from Apalachia Dam. The overall population along the Hiwassee River does not appear to be at risk of extirpation, but some sites could be lost over time under current flow management guidelines.

From November 2015 until March 2016, TVA performed scheduled maintenance on the Apalachia powerhouse and switchyard, which is situated along the Hiwassee River. This work necessitated the closure of the powerhouse and diversion of all flows through the “dry” section of the Hiwassee River where *P. ruthii* occurs. Typically, flow in this section of the river consists only of base releases of 25 cfs from Apalachia Dam plus tributary inputs, which are driven by local rainfall. Therefore, under normal operating conditions, higher flows in this section of the river are intermittent and short-term in duration. During the maintenance work, flows were elevated for 132 days. Several high rainfall events also occurred during this time, which necessitated higher-than-planned releases from Appalachia Dam. Daily averages varied, but releases ranging from 2500 to 5000 cfs were common; the highest and lowest releases during that time period were 8386 and 541 cfs, respectively. At some sites, individual *P. ruthii* plants were inundated for nearly all of this period, whereas other sites where *P. ruthii* occurs well above the river channel were not inundated. Census counts conducted in the fall of 2016 indicated a decrease in plants on the Hiwassee River, whereas counts on the Ocoee River remained stable. This suggested that population declines observed along the Hiwassee River were related to the elevated flows present during the Apalachia powerhouse maintenance period rather than the simultaneously occurring drought conditions that affected both watersheds during 2016.

Releasing periodic higher flows has been cited as an action that could reduce woody plant encroachment that degrades *P. ruthii* habitat along the Hiwassee River, but observations of occupied habitat along the river during and after the maintenance on Apalachia powerhouse suggested that managed high releases alone are not likely to be effective at reversing this threat. The magnitude and duration of this flow event was unparalleled since Apalachia Dam was completed in 1943. These historically high flows did not appreciably change the extent of woody vegetation along portions of the Hiwassee River supporting *P. ruthii*. Monitoring data do suggest periodic elevated flows might stimulate population increases at currently occupied sites, presumably through increased recruitment and establishment of seedlings, but it is unlikely that releases from Apalachia Dam could reduce established woody plant encroachment on a meaningful scale. Factors that limit the effectiveness of this approach include both operational and ecological constraints. The presence of downstream homes and businesses in the Hiwassee River watershed places an upper limit on releases that can be sustained, even for short periods, without increasing the risk of harm to life and property due to flooding. This operational constraint is compounded by the alteration, over time, of the riparian vegetation community along the river. The fact that mature woody vegetation has become well established in many parts of the narrow riparian zone where phyllite outcrops would otherwise be available for the recruitment and establishment of *P. ruthii* presents an ecological constraint. Alteration of flows since the late 1940s has resulted in the establishment of vegetation that is resistant to low-frequency, moderately high flows that are operationally possible due to downstream flooding concerns.

Recreational flows have likely helped to maintain the habitat for *P. ruthii* and increase the population size along the Ocoee River, although there is no way to prove causality from the correlation between increased periodic flow events and the overall population increase. A methodology used on the Youghiogheny River in Pennsylvania to look at the effects of river flow on another species in the Asteraceae, *Marshallia pulchra* W.M. Knapp, D.B. Poind & Weakley [47] could be used to better understand the relationship between flows and inundation of *P. ruthii*’s habitat on the Ocoee River and inform flow management for the Hiwassee River. Examining the frequency and duration of inundation events along the Ocoee River, which has stable or increasing population numbers, and then using that information to assess flow regimes at a mix of stable and declining sites on the Hiwassee River may offer important guidance on how possible future changes to flows from Apalachia Dam may affect the species. It is possible that flow management, combined with periodic drought, could provide habitat conditions sufficient and conducive to maintaining demographic structure and genetic variation within the subpopulations now recognized in the Hiwassee River, such that they are self-sustaining. However, additional management to reduce woody vegetation encroachment could be needed in some sites to restore open conditions where flow alterations over the past eight decades have allowed woody riparian vegetation to become established to an extent that will be irreversible through flow alteration alone. In addition, given the susceptibility *P. ruthii* to extended periods of inundation, any proposed changes to the 25 cfs base flow in the Hiwassee River would need to be carefully considered to ensure the increase would not negatively impact the species, particularly clusters of plants that occur at lower elevation near the current active channel.

Because of the very restricted range and evident lack of competitiveness in suitable habitats, it is obvious that any change in the home habitat, either minor or major, of *P. ruthiii* would probably negatively affect the survival of this plant. Climate changes as manifested in temperature changes, and especially the amount of rainfall could possibly influence the recovery and stabilization of the existing populations. Increased or decreased rainfall and the resultant fluctuations of water flow on the two home river systems of *P. ruthii* could potentially alter habitat characteristics and seed distribution and limit the total area available to support and establish populations of Ruth’s Golden Aster. The number of plants and genetic diversity of *P. ruthii* subpopulations must be monitored, cataloged frequently, and correlated with local climatic variables over time to assess changes in population structure and survival of this unique species.

Furthermore, the reasons why *P. ruthii* is rare are still unknown. If the answers to these questions about rarity were known, arguably, we would be in a much better position to fully recover the species. However, the factors thought to be responsible are (1) association with a specific, not widely available geologic substrate; (2) dependence on a now-gone hydrologic regime to (a) provide sufficient influx of sediment to support plant establishment in the crevice habitats the species occupies and (b) regulate populations of other herbaceous taxa competing for resources in those crevices or woody/vine taxa that cast excessive shade and limit growth/reproductive output; and (3) variable production of filled seeds that are capable of germination and successfully establishing new plants in a taxing environment that likely limits the number of individuals able to transition from seedling to later life history stages, presumably due to inbreeding as a result of low numbers of compatible mates. Boyd et al. [29] attempted to shed some light on this question of rarity through ecophysiological and genetic comparisons with the more common *P. graminifolia* var. *latifolia*.

The purpose of this review is to highlight the relevant research efforts that have delineated the range and size of populations, determined habitat requirements, documented reproductive and propagation potential, and comprehend the demographic, ecological, and genetic factors that may increase vulnerability to extinction for *P. ruthii*. Cooperative partnerships have led to the successful completion of specific actions called for in the species recovery plan for *P. ruthii*. These partnerships have been invaluable in filling knowledge gaps and providing a foundation for the USFWS to guide conservation and management decisions for the species. A major lesson learned is that consistent effort/attention over time to recovery efforts for a given species is difficult to maintain, but crucial for maintaining momentum. Maintaining regular communication among recovery partners is key, and the recovery work for *P. ruthii* reflects what has been successful for other species as well: when the partners are working together with effective communication, progress happens more quickly. Whether led by the USFWS or by other engaged partners, effective communication is needed to make recovery happen.

## Figures and Tables

**Figure 1 plants-12-02693-f001:**
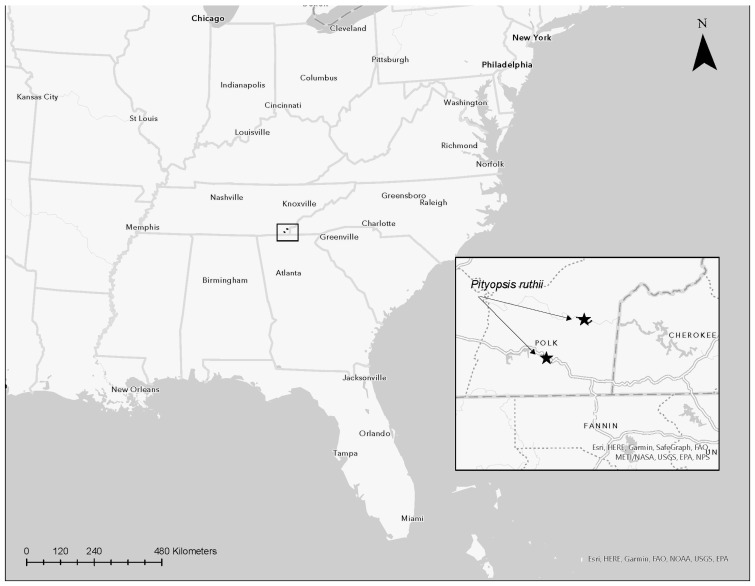
Distribution of the endangered *Pityopsis ruthii* (Ruth’s golden aster) on the Hiwassee and Ocoee rivers in Southeastern Tennessee, United States of America. See Hatmaker et al. [13] for detailed information regarding the distribution of subpopulations and population structure for the Hiwassee and Ocoee River populations.

**Figure 2 plants-12-02693-f002:**
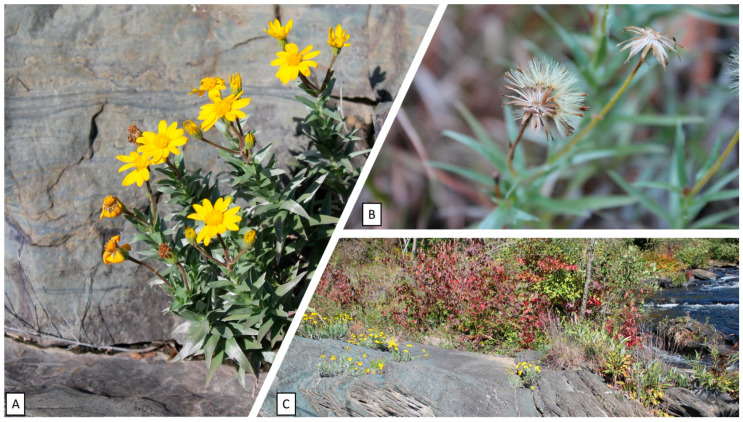
The endangered *Pityopsis ruthii* (Ruth’s golden aster) growing along the riparian corridor on the Hiwassee River in Southeastern Tennessee, United States of America. (**A**) An individual plant exhibiting erect to ascending habit with lanceolate leaves that are silvery pubescent and overlap in tight spirals. Inflorescences are either solitary or in a cyme (1.5–2 cm diameter heads), composed of 8–15 ray florets, with bright yellow petals and numerous disk florets; (**B**) the florets produce ~4 mm long pubescent achenes each with a 4–5 mm long pappus; (**C**) the herbaceous perennial is characterized by habitat specialization, as distribution is restricted to discrete locations on exposed phyllite boulders within and along the banks of the Hiwassee and Ocoee rivers.

**Table 1 plants-12-02693-t001:** Summary of recovery plan implementation progress for *Pityopsis ruthii*.

Recovery Action	Description	Action Status *
1	Maintain formal agreements with agencies concerned with the preservation of *P. ruthii*	Not started
2	Maintain permanent plots	Complete
3	Determine effective and successful achene dispersal, seed germination, and seedling establishment	See subactivities
3.1	Study achene dispersal	Discontinued
3.2	Determine life history, seed germination, and seedling establishment requirements	Discontinued
3.3	Determine the role of inter- and intraspecific competition	Not started
4	Determine what constitutes suitable habitat	Discontinued
5	Search for *P. ruthii* on other rivers	Completed
6	Determine and implement management for the Hiwassee River population for long-term reproduction, maintenance, and vigor	See subactivities
6.1	Determine and compare past and present stream flow regimes	Ongoing not current
6.2	Determine the nature and role of natural succession in habitat	Not started
6.3	Determine if the population is self-sustaining	Ongoing current
6.4	Establish *P. ruthii* in a suitable unoccupied habitat	Discontinued
6.5	Establish a cultivated population from Hiwassee River and provide long-term seed storage	Ongoing current
6.6	Determine feasibility and/or necessity of water releases and hand-clearing of phyllite boulders	Partially complete
7	Determine and implement management for the Ocoee River population for long-term reproduction, maintenance, and vigor	See subactivities
7.1	Study the relationship of the river to *P. ruthii*	Partially complete
7.2	Determine impacts on river recreational users and implement management actions	Ongoing current
7.3	Ensure that highway construction will not damage or destroy plants or suitable habitat	Ongoing current
7.4	Determine if the population is self-sustaining	Ongoing current
7.5	Establish *P. ruthii* in a suitable unoccupied habitat	Completed
7.6	Establish a cultivated population from Ocoee River and provide long-term seed storage	Ongoing current

* Action status definitions are from the U.S. Fish & Wildlife Service Environmental Online System (FWS ECOS). Discontinued = action has had some work done but is out-of-date or unsuccessful. Still considered necessary for recovery, but there are no current plans to resume work. Complete = action has been successfully completed. No work remains to be done. Not Started = no planning or implementation work has been carried out. No plans in place to begin work. Still considered necessary for recovery. Obsolete = this action is not necessary for recovery according to the current understanding of the species’ status. Ongoing current = action duration is ‘ongoing’ or ‘continuous’ (i.e., actions without specified endpoints that are conducted continuously or periodically throughout the recovery process, like surveys). Action is considered necessary for recovery and is currently being successfully implemented. Further work is needed to bring the action to completion. Ongoing not current = action duration is “ongoing” or “continuous” (i.e., actions without specified endpoints that are conducted continuously or periodically throughout the recovery process, like surveys). Action is still considered necessary for recovery, but is behind schedule (not current). Partially complete = action duration has a discrete endpoint (i.e., 3 years). Action has been partially completed (relative to the work needed when the recovery plan was released). Planned = initial planning of action is complete or in progress, but no implementation has yet been done (relative to work needed when the recovery plan was released). Unknown = status of action planning or implementation not known.

## Data Availability

Not applicable.

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
