# Peer review of "Pityopsis ruthii: An Updated Review of Conservation Efforts for an Endangered Plant"

_plants, 2023, doi:10.3390/plants12142693_

Round 1
Reviewer 1 Report
This is a useful review of decades of work on a very restricted and habitat specific plant, some of which is in government reports and graduate theses that are not readily assessible. There are a few things the authors should consider to improve the manuscript.
Section 1. Introduction. There is discussion here of the impacts of climate change on plant survival, and this is an important consideration. However, this never really comes into the discussion of the future survival of this species. I suggest some tie if you want to retain climatic change in the introduction. A couple things might be considered. Climate predictions generally indicate greater variability in future climates. Droughts might benefit this species by limiting hardwood establishment and growth. More flooding from episodes of high precipitation (to the extent that it would occur in a regulated system) might also help this species. I do not know if there is any information on the temperature tolerances of the plant.
Author line. There appears to be an extraneous "and *" at the end of the line.
Reference #37. Is there a date for the Wofford and Smith report?
Reviewer 2 Report
Upon first reading the abstract, I felt it did not say much. Maybe it might be useful to include some of the contents of the review in the abstract? Just a brief summary, perhaps, of how previous research has led to newer approaches, and how the survival of this species is linked to water management, might foster greater interest in the paper.
The manuscript itself read quite well, though I have suggested a few re-wordings and insertion of commas to help the comprehension of long sentences. I attach the word document that I commented on using 'track changes'.
Round 2
Reviewer 1 Report
Minor correction.
Page 3, par. 5. Italicize latifolia in P. graminifolia (Michx.) Nutt. var. latifolia (Fernald) Semple & F.D. Bowers.
Reviewer 2 Report
I still see a highlighted area on p. 3 where there is some italics needed (for the latin name only)